# Cryotherapy in the Treatment of Extra-Abdominal Desmoid Tumors—A Review

**DOI:** 10.3390/curroncol32030137

**Published:** 2025-02-27

**Authors:** Kadhim Taqi, Cecily Stockley, Melissa Wood, Stefan Przybojewski, Antoine Bouchard-Fortier, Lloyd Mack

**Affiliations:** 1Division of Surgical Oncology, Department of Oncology, Cumming School of Medicine, University of Calgary, Calgary, AB T2N1N4, Canada; kadhim.taqi@ucalgary.ca (K.T.); cecily.stockley@albertahealthservices.ca (C.S.); melissa.wood@albertahealthservices.ca (M.W.); antoine.bouchard-fortier@albertahealthservices.ca (A.B.-F.); 2Department of Radiology, Cumming School of Medicine, University of Calgary, Calgary, AB T2N1N4, Canada; stefan.przybojewski@albertahealthservices.ca

**Keywords:** desmoid fibromatosis, desmoid tumor, cryoablation, cryotherapy, ablation

## Abstract

The management of desmoid tumors (DTs) often poses challenges due to their variable clinical behavior, with treatment options including active surveillance, systemic therapy, and local therapies including surgery, ablation, and radiation. More recently, cryotherapy has emerged as a promising localized treatment for DTs. We aimed to conduct a review of the indications, techniques, and outcomes of cryotherapy in the treatment of extra-abdominal DTs. The review suggests that cryotherapy can be effectively used for both curative and debulking purposes, with a significant number of patients achieving symptom relief, disease stabilization, or regression. Although generally safe, cryotherapy is associated with potential risks, particularly when critical structures are in proximity. Overall, cryotherapy offers a viable, minimally invasive treatment option for DTs, with favorable outcomes in both symptom relief and tumor control.

## 1. Introduction

Desmoid tumors (DTs) are rare, connective tissue neoplasms which have no distant metastatic potential, but can be both locally aggressive and invasive [1,2,3,4]. DTs pose an ongoing dilemma to healthcare providers in their management due to their variable behavior and their impact on both quality of life and activities of daily living for a proportion of patients [1,2]. The vast majority of DTs arise sporadically, but they can also arise due to trauma (e.g., surgery), pregnancy, and in some individuals who have a genetic predisposition with an Adenomatous Polyposis Coli (*APC*) gene mutation [5,6]. Most sporadic tumors are associated with a mutation in exon 3 in the Catenin beta 1 (*CTNNB1*) gene [4].

The variable behavior of DTs makes management complex; approximately 20–28% will regress spontaneously, 32–60% will remain stable, and 20–40% will progress [7,8]. Treatment is generally initiated for both growth or progression and symptoms [8]. In the recently published global consensus from the Desmoid Tumor Working Group, it is recommended that active treatment be delayed until the progression or increase in symptom burden can be assessed in at least two subsequent assessments, unless the tumor is close to critical structures where further progression could be life-threatening, as can be seen in head and neck and mesenteric desmoids [9]. If active surveillance fails and active treatment is required, management options include localized treatments and systemic treatments. Localized modalities include surgery, radiation, and ablative methods, including cryoablation, microwave ablation (MWA), radiofrequency ablation (RFA), trans-arterial chemoembolization (TACE), and high-intensity focused ultrasound (HIFU) therapy [10,11].

Cryotherapy is an interventional procedure which employs repeated cycles of freezing and thawing with the purpose of causing cell death [12]. It has recently emerged as a novel localized treatment modality for DTs. There have been several retrospective and two prospective studies examining the efficacy and safety of cryoablation; however, most of these studies report single-center heterogeneous experiences. The primary objective of this review was to discuss the role of cryotherapy in the management of DTs, and to provide a detailed understanding of how cryotherapy fits within the broader spectrum of DT management strategies and its potential benefits and limitations.

## 2. Indications for Cryotherapy in Desmoid Tumors

Cryoablation is utilized as either a first-line therapy or a salvage therapy in DTs following the failure of other treatment modalities, ideally within the context of a multidisciplinary tumor board review. It may be performed with curative intent or for debulking (partial tumor destruction) to achieve symptom relief [13]. Currently, the procedure seems to be more commonly employed as a second- or third-line treatment (39–100%) following local or systemic therapies [14,15]. Cryoablation is considered curative if the entire tumor volume can be effectively targeted without posing a significant risk to adjacent neurovascular, digestive, or urinary structures. When such risks limit the extent of the procedure, it is classified as debulking [13,16]. It has been suggested that targeting at least 50% of the tumor volume is necessary to provide clinical benefit [13]. Consequently, cryoablation may be repeated based on the clinical and radiological response, and this can be performed multiple times if necessary [14,16,17,18,19,20,21].

There are no established tumor size criteria in the literature that would preclude patients from undergoing cryoablation. However, reported cases have described tumor sizes ranging from 8.83 mm to 209 mm [14,22] and tumor volumes between 0.35 cm^3^ and 1006 cm^3^ [14,21]. Common anatomical locations for the cryoablation of DTs include the trunk, extremities, abdominal wall, chest wall, paraspinal region, gluteal/hip area, and head and neck [13,14,15,16,17,18,19,20,21,22,23,24,25,26,27].

## 3. Cryotherapy Techniques

Cryoablation employs the use of cryoprobes to deliver freezing temperatures to a tumor, inducing irreversible damage [28]. Cryoprobes utilize the Joule–Thomson effect, where the rapid expansion of liquid gas (often argon) leads to a temperature drop and subsequent cooling along the probe [28,29]. The procedure involves freeze–thaw cycles [28] (Figure 1). An ice ball forms around the tumor, inducing cell death. Subsequently, there is a thawing phase, followed by another freezing phase. Tissue damage is induced directly and indirectly. Typically, there are two cycles of 10-min freezing, followed by 5-min thawing [14,16,18,29]. Some studies describe 10-min thawing in between freezing cycles [13,29]. Bergin et al. describe 10-min freezing, followed by 5-min passive thawing (naturally, without the application of external heat), and another 10-min freezing, followed by 5-min active thawing (with the application of heat, typically helium or electricity) [21,30]. All studies describe that freezing can be shortened if the ice ball appears to be too close in proximity to critical structures [20,21,23].

During the freezing phase, there is intra- and extra-cellular formation of ice crystals, causing direct injury. During thawing, there is vascular stasis and tissue ischemia. Post-procedure, for up to 8 h, there is cell death by apoptosis due to an osmotic effect of the thawing ice ball. Lastly, there is inflammation and immunologic response through dendritic cells that cause late and continued effects of cryoablation [28]. Al-Assam et al. recommend extending the ice ball 5mm beyond the tumor, as the peripheral 5mm does not cause cell death [29].

Pre-procedure planning involves diagnostic imaging, either computed tomography (CT) or magnetic resonance imaging (MRI), to measure the tumor volume and location relative to critical structures [14,20]. One center described using MRI to create 3D models to plan probe placement [31]. Tumor volume is estimated by the formula ½ × (length × width × height) and includes both viable and previously ablated non-enhancing tissue [14]. Contrast-enhanced ultrasonography has also been described in the literature, as it often demonstrates a typical pattern of prolonged wash-out, which is characteristic of benign lesions. It can serve as an alternative to CT and MRI [31].

At the time of the procedure, patients undergo general anesthesia, conscious sedation, or local anesthesia [16,21]. Locoregional blocks may also be performed for post-procedure pain [13]. The procedure then begins by placing probes in the tumor based on predefined locations from imaging and the relationship of the tumor to critical structures [14,18,20,29]. The number of probes typically depends on tumor volume, with one to fifteen 15-gauge probes placed in the lesion about 1.5 cm apart in all tissue planes under imaging guidance with confirmation on CT or fluoroscopy prior to procedure initiation [13,16,21] (Figure 2a,b).

Diagnostic imaging, often with CT, is typically performed again at the end of the procedure to define the maximum ablation zone and assess for any injuries to critical structures or bleeding [14]. Post-procedure, a common assessment of remaining viable tumor is known as the A-status: A0 represents complete ablation with no further enhancement, and A2 represents residual enhancement, with A2a having ≤15% residual disease and A2b > 15% residual disease [14].

Depending on the complexity of the procedure, most patients are often discharged the same day, with the average length of stay in the literature ranging between 0 and 4 days [14,22]. Patients are admitted often for monitoring, for post-procedural symptom control (for pain or nausea), or in the event of a complication [14].

## 4. Cryotherapy Safety and Side Effects

Tissues at risk of injury during cryoablation are skin, neurovascular, digestive, or urinary structures [13,16,21,28]. In addition to decreasing the freeze time if the ice ball is approaching critical structures, precautionary measures are taken during the procedure to reduce the risk of injury to surrounding structures if these structures are within 0.5–1 cm of the ice ball [16,21]. Techniques include hydro-dissection between the tumor and nearby structures, as well as between the tumor and skin, passive skin warming with warm saline in a sterile glove placed on the skin, and nerve monitoring as needed [14,16,21] (Figure 3). Another technique described is dynamic hydro-dissection, which entails the continuous monitoring and adjustment of subcutaneous hydro-dissection throughout the cryoablation procedure. This approach ensures that a safe distance of greater than 5 mm is consistently maintained between the ice ball and the skin during the procedure [32]. Bergin et al. also describe giving intra-procedural dexamethasone to decrease post-procedure nausea and inflammation, as well as a 5-day course of dexamethasone to reduce the risk of compartment syndrome for large DTs of the extremities [21].

The complication rate varies across centers, ranging from 0% to 79% [22,26]. The majority of reported complications are classified as grade 1 or 2, whereas the incidence of grade 3 or 4 complications ranges from 0% to 30% [17,22,26,27]. Commonly reported adverse events include paresthesia, skin injury, pain, nerve injury, rhabdomyolysis, swelling, hematoma, bleeding, emesis, and pleural effusion [14,15,16,17,19,21]. Less frequently observed complications include weakness, muscle necrosis, restricted range of motion, pneumothorax, wound infection, brachial plexus injury, bruising, colo-cutaneous fistula, and acute kidney injury [18,20,21,22,25,27]. The most commonly affected nerves include the peroneal, radial, spinal accessory, brachial plexus, and tibial nerves, and in majority of the cases, the injury is transient [13,14,15,16,19,21,23,25].

## 5. Symptom Relief

Symptomatology remains one of the main indications for DT treatment with cryoablation. Early experiences suggest that the treatment can be effective in relieving pain and reducing tumor size [33]. One of the first systematic reviews assessing the role of cryoablation in extra-abdominal desmoid management, with 146 patients across studies, showed that complete pain relief was reported in 40% to 66.7% of patients [31]. Another systematic review involving 214 patients across nine studies showed that 37.5% to 96.9% of the patients reported having experienced partial or complete symptom relief following cryoablation [34]. Improvement in both symptom intensity and effect on activities of daily living occurs gradually, typically over a period of approximately 6 months following the procedure [16,27]. While pain remains the main symptom assessed in the literature, other symptoms reported to improve following ablative therapy include motor dysfunction, abdominal distension, and abdominal pressure [33]. Most of the studies in the literature, however, assess response subjectively, with few studies assessing symptom relief with quantitative scales [33]. Bouhamama et al., in a retrospective review of 34 patients, used a visual analog score (VAS) to measure pain intensity and response to treatment at two points: pre-treatment and six months following treatment. They showed that following cryoablation, the VAS decreased by 3.3 points, which was statistically significant (VAS pre-treatment of 5.7 versus VAS at six months post cryoablation of 2.4, *p* < 0.001) [35].

The recurrence of symptoms following treatment has been reported to occur in approximately 37.5% of patients, with a median time for symptom recurrence of around 10 months [16]. A portion of these patients requires a second procedure, which generally takes place after the six-month follow-up period. On average, the time between the initial procedure and the repeat procedure is approximately 10 months [14,16]. A recent systematic review with meta-analysis showed no statistically significant difference in symptom relief between cryoablation and other local therapies including MWA and HIFU, (pooled symptom relief rate for cryoablation—87%; MWA—89%; HIFU—100%; *p* = 0.249). Most of the symptoms did not re-occur on follow-up [33].

## 6. Diagnostic Imaging Assessment of Treatment Response

Following cryoablation, MRI appears to be the preferred imaging modality for monitoring treatment response; however, interpreting the changes in DT signals on imaging may be difficult, and there is limited research on the possible expected changes [36]. This challenge stems from the freezing cycle used in cryotherapy, which induces apoptosis. As a result, solid tumors such as DTs can transform into gelatinous necrosis that resorbs slowly [36]. Studies on kidney models have shown an increase in high-diffuse-weight image (DWI) signals in the first months following cryoablation, likely due to edema and swelling, followed by hypointense signals in the following six to nine months, likely due to scar formation [36]. At present, there is no universally accepted method for describing and evaluating objective tumor response after cryoablation. Methods employed include assessments of the total lesion volume (TLV), viable tumor volume (VTV), non-perfused volume rate (NPVR), and the RECIST or mRECIST criteria (Table 1) [16,37].

Studies have shown that TLV tends to increase in the immediate phase (0–3 months) due to tissue swelling by around 18.2%, followed by a decrease of around 6.7–36.7% from the original pre-procedure TLV [16,18]. The challenge with using TLV alone in assessing treatment response is that TLV can also increase up to 20–40% by 12 months; however, this does not necessarily translate into a clinical recurrence or treatment failure, as it could represent a non-viable tumor [23,31]. The VTV method, in contrast, reveals more pronounced changes, with an approximate reduction of 56% in the first phase (0–3 months) and a final reduction of about 43–87% by 6–12 months compared to the pre-procedure VTV [16,18,23].

The NPVR measures the ratio of the ablated area to the total tumor volume immediately after ablation and uses contrast-enhanced T1-weighted (T1W) imaging. This method has also been applied to other solid lesions, such as fibroids [38]. In their systematic review, Huang et al. showed that NPVR changes ranged between 65.2 and 92.5% across 23 studies, with a pooled NPVR change of 76%. This was reflected clinically, with most patients experiencing clinical benefits from changes in the viable spatial dimensions of the tumor [33].

The RECIST or mRECIST employs both the TLV and the VTV to classify results into complete response (CR), partial response (PR), stable disease (SD), or progressive disease (PD) and appears to be more commonly used in the literature [37,39] (Table 1). When using mRECIST criteria to assess tumor response by 12 months, studies show that the CR rate ranges between 0% and 43.3%, the partial response (PR) rate from 26.2% to 83.3%, the stable disease (SD) rate from 0% to 43%, and the progressive disease (PD) rate from 0% to 31.8% [15,16,21,22,26].

## 7. Assessing Clinical Outcomes

Maintaining disease stability and ensuring long-term disease control are crucial components of DT treatment. The main two reported outcomes in the literature are progression-free survival (PFS), which is the time from randomization or treatment to disease progression or death, and disease-free survival (DFS), which is the time from treatment to disease recurrence or death from any cause, typically used after curative-intent local therapy [40]. PFS is utilized more to assess the efficacy of treatments in delaying disease progression, particularly in cancers that are not curable but can be controlled. DFS is typically used when the goal is to measure how long patients stay disease-free after initial treatment. PFS, unlike DFS, can also represent stable but not necessarily eradicated disease [40]. For these reasons, PFS may be more representative of tumor response following cryoablation. The 1-year PFS following therapy reported in the literature ranges between 85.1 and 89%, while the 3-year PFS is approximately 68–77.3%. The 1-year DFS following cryotherapy ranges between 21.7 and 62%, while the 2-year DFS is around 82.3% [13,14,15,16,22,23,24,25,27]. Figure 4 and Table 2 summarize these findings.

## 8. Follow-Up

Post-cryoablation follow-up criteria are not well defined, and most institutions base their surveillance protocols on extrapolations from established active surveillance practices, incorporating both clinical and radiological evaluations [40,41,42]. Follow-up intervals are typically set at 2 to 6 months, with imaging assessments often including CT scans, MRIs, or a combination of both modalities [41]. Most institutions transition to annual surveillance if disease stability is maintained for 2–3 years, with this surveillance extending up to approximately 5 years [40,41]. Following cryoablation, the majority of studies report follow-up durations ranging from 3.6 to 53.7 months [16,33], with follow-up appointments every 3 months during the first year [43]. In their prospective trial, Kurtz et al. followed their patients for 12 months following cryoablation, with clinical examination and MRI every 3 months [27]. Given the natural history of DTs and the reported 3-year PFS rates of 68–77.3% and 2-year DFS rate of 82.3% following cryotherapy [15,16,23], monitoring patients more frequently during the first 3 years with extended follow-up for 5 years post-treatment with annual CT or MRI is advisable.

## 9. Conclusions

Cryotherapy has emerged as a valuable treatment modality for the management of extra-abdominal DTs, particularly in patients for whom surgery is not an appropriate option or previous treatments have proven inadequate. It can effectively stabilize or reduce tumor size, offering both symptomatic relief and disease control. While the procedure is generally well tolerated, careful consideration must be given to the proximity of critical structures to minimize risks. As our understanding of DTs continues to evolve, cryotherapy provides a minimally invasive option that complements existing therapies. However, further research is essential to establishing standardized protocols, improving patient selection, and optimizing outcomes, thereby enhancing the role of cryotherapy in the multidisciplinary management of DTs.

## Figures and Tables

**Figure 1 curroncol-32-00137-f001:**
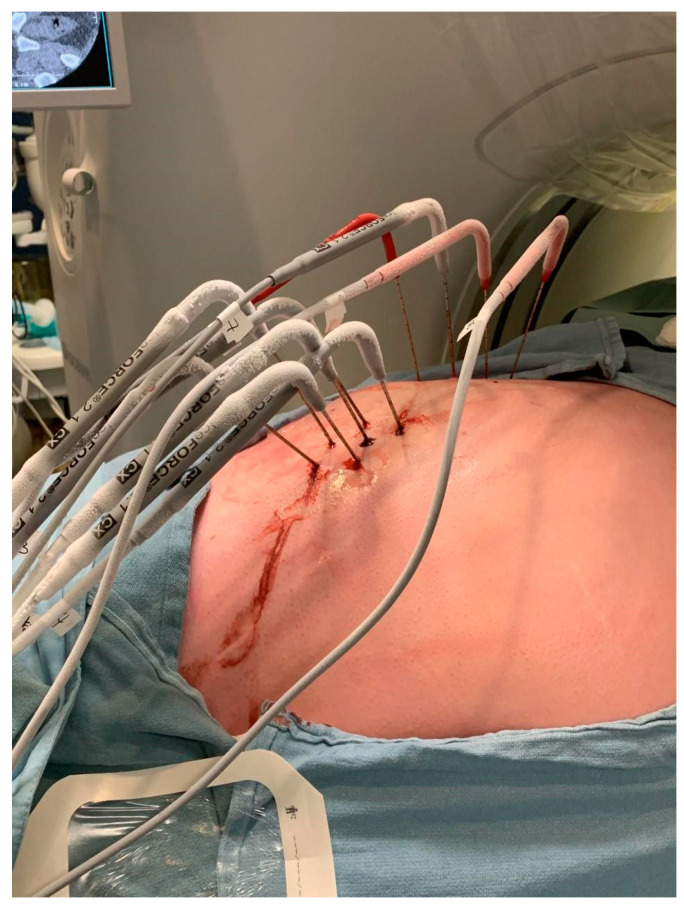
Probe placement is performed under image guidance. This image shows an active freeze cycle (ice accumulation on the probes) during percutaneous cryoablation.

**Figure 2 curroncol-32-00137-f002:**
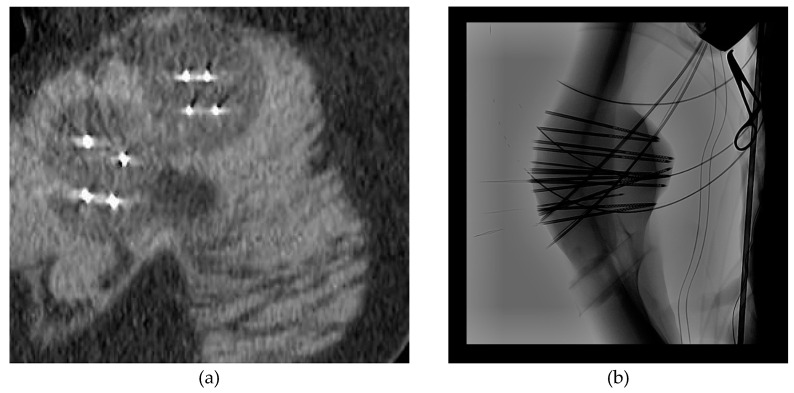
Computed tomography scan (**a**) or fluoroscopy (**b**) is performed prior to initiation of freeze cycles to confirm probe placement and assess surrounding structures.

**Figure 3 curroncol-32-00137-f003:**
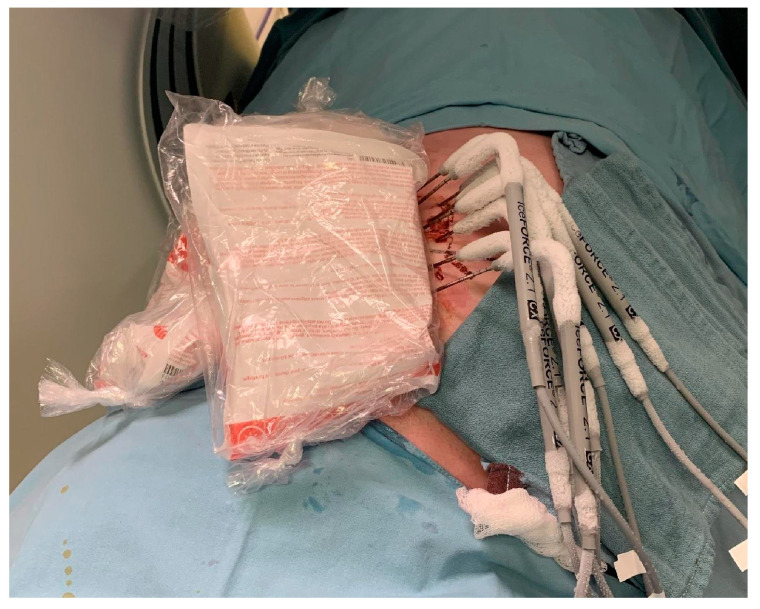
Thaw cycles can be active or passive. In this image, sterile heat pads are applied to help the ice ball melt.

**Figure 4 curroncol-32-00137-f004:**
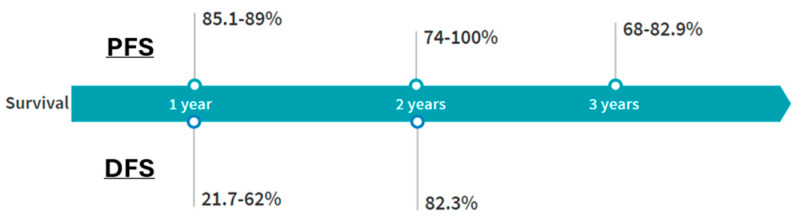
Progression-free survival (PFS) and disease-free survival (DFS) following cryotherapy in extra-abdominal desmoid tumors in the literature.

**Table 1 curroncol-32-00137-t001:** Modified Response Evaluation Criteria In Solid Tumors (mRECIST) criteria in assessing desmoid tumors’ response to treatment [37].

Response	Definition
Complete response (CD)	Disappearance of contrast enhancement of all lesions.
Partial response (PR)	≥30% decrease in the sum of the diameters of contrast enhancement of target lesions.
Stable disease (SD)	Neither partial response nor progressive disease.
Progressive disease (PD)	≥20% size increase in contrast enhancement of the target lesions or new disease.

**Table 2 curroncol-32-00137-t002:** Studies in the literature reporting survival outcomes including progression-free survival (PFS) and disease-free survival (DFS), and the clinical and radiological responses in these studies.

Study	Mean or Median Follow-Up (Months)	Response (Tumor *n*, %) mRECIST Criteria	Symptom Relief (Patient *n*, %)	PFS	DFS
Havez et al., 2014 [23]	7	CR: 1, 5.8 PR: 7, 41.3 SD: 7, 41.3 PD: 2, 11.6	13, 100	-	2 years: 82.3%
Tremblay et al., 2019 [14]	15.4	CR: 8, 36 PR: 8, 36 SD: 7, 28 PD: 0, 0	18, 90	-	1 year: 21.7%
Bruyns et al., 2020 [24]	77	CR: 1, 20 PR: 3, 60 SD: 1, 20 PD: 0, 0	-	2 years: 100%	-
Saltiel et al., 2020 [25]	53.7	CR: 4, 40 PR: 5, 50 SD: 3, 30 PD: 2, 20	2, 20	-	3 months: 90% 6 months: 62% 1 year: 62%
Auloge et al., 2021 [15]	18.5	CR: 13, 43.3 PR: 11, 36.7 SD: 1, 3.3 PD: 5, 16.7	29, 96.7	1 year: 85.1% 3 years: 77.3%	-
Kurtz et al., 2021 [22]	31	CR: 12, 28.6 PR: 11, 26.2 SD: 13, 31 PD: -	-	1 year: 85.8%	-
Yan et al., 2021 [16]	10	CR: 0, 0 PR: 8, 61.5 SD: 4, 30.8 PD: 1, 7.7 α	32, 96.9	3 years: 82.9%	-
Mandel et al., 2022 [27]	16.3	CR, PR, SD: 15, 68.2 PD: 7, 31.8	14, 63.6	2 years: 59% *	-
Bouhamama et al., 2023 [13]	23.8	-	55, 83.3	1 year: 89% 2 years: 74% 3 years: 68%	

CR: complete response. PR: partial response. SD: stable disease. PD: progressive disease. PFS: progression-free survival. DFS: disease-free survival. * Local recurrence-free survival (LRFS). α Patients with complete ablation only.

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
