# Peer review of "Cryotherapy in the Treatment of Extra-Abdominal Desmoid Tumors—A Review"

_curroncol, 2025, doi:10.3390/curroncol32030137_

Round 1

Reviewer 1 Report

Comments and Suggestions for Authors

Title

1. Appropriate

Abstract

2. "Further research is needed to refine patient selection criteria and optimize treatment protocols." Strike or save for the conclusion.

Introduction

3. "TACE has been used successfully with doxorubicin drug-eluting beads, with reported reductions in tumor volumes of 50-97%, and most patients reported symptomatic improvement [27,30]. Consider adding reference for Elnekave et al, Selective Intra-Arterial Doxorubicin Eluting Microsphere Embolization for Desmoid Fibromatosis: A Combined Prospective and Retrospective Study, Cancers (Basel) 2022 Oct 14;14(20):5045. doi: 10.3390/cancers14205045.

4. "Systemic treatments used to treat DTs include nonsteroidal anti-inflammatory drugs (NSAIDs), antihormonal therapies, tyrosine kinase inhibitors (TKIs, such as Sorafenib and Pazopanib), gamma secretase inhibitors (Nirogacestat and AL102) and cytotoxic chemotherapy [4,24]. " Please expand citations for this sentence, optimally citing primary literature, including, but not limited to:

Constantinidou et al, Pegylated liposomal doxorubicin, an effective, well-tolerated treatment for refractory aggressive fibromatosis. 2009 European Journal of Cancer , 45(17), pp. 2930–2934

Gounder et al, Nirogacestat, a γ-Secretase Inhibitor for Desmoid Tumors. 2023 New England Journal of Medicine, 388(10), pp. 898–912

5. "Cryotherapy is an interventional procedure which employs repeated cycles of freezing and thawing with the purpose of causing cell death [33]."  Citation 33 is not appropriate here, as citation 33 is a clinic study. Consider. Erinjeri et al Cryoablation: mechanism of action and devices. J Vasc Interv Radiol. 2010 Aug;21(8 Suppl):S187-91. doi: 10.1016/j.jvir.2009.12.403

Material and Methods

6. Appropriate

Results

7. Appropriate

Discussion

8. "Techniques include hydro-dissection between the tumor and nearby 254 structures, as well as between the tumor and skin, passive skin warming with warm saline 255 in a sterile glove placed on the skin, and nerve monitoring as needed". Consider this reference, not for desmoid, but shows a nice example directly: Sanderg et al, Dynamic Hydrodissection for Skin Protection during Cryoablation of Superficial Lesions. J Vasc Interv Radiol. 2020 Nov;31(11):1942-1945. doi: 10.1016/j.jvir.2020.01.025. Epub 2020 May 14.

Conclusion

9. Appropriate

Tables

10. In Table 1 "Mean Tumor size (range)mm", Tumor should be lower case. 

Reviewer 2 Report

Comments and Suggestions for Authors

A very well-written review on the use of cryoablation for the treatment of desmoid-type aggressive fibromatosis, which offers a good overview of the field . Since the use of ablative techniques has expanded, and each one has advantages and disadvantages, I suggest that the authors add a comparison between cryotherapy, high-intensity focused ultrasound, microwave- and radiofrequency ablation in sections 4.3, 4.5 and 4.6 similarly to their last paragraph in 4.5 (and if possible in a separate Table, since this would be useful for physicians treating this condition). 

Reviewer 3 Report

Comments and Suggestions for Authors

I appreciated the review.

I have several suggestions to improve the paper:

1) TITLE: "Systematic" instead of "Scoping" may help in highlighting the great potential of this article and its comprehensive content.

2) INTRODUCTION: Imaging/Radiological section is complitely missing. Please provide a dedicated section including MRI classic appearence above all, and possibly ultrasound (sugg. ref. PMID: 22109314); Include also a brief paragraph on imaging differential diagnosis, the main ones are soft-tissue lymphomas (sugg ref. PMID: 36088202) that may be in differential with desmoid tumors in their very "active" phase.

3) METHODS: If this is a "systematic" review article, the criteria of inclusion/exclusion and the search strategies should be better specified in methods section. Particularly, adherence to PRISMA guidelines for systematic reviews is strongly suggested.

4) FIGURES: Key-radiologic images are suggested in the paper. Figure 3 and Figure 4 are very similar. I suggest You to replace one of them with the radiological real-time control (e.g. CT 'ice-ball' intra-procedural) or adding a couple of figures (MRI or CT). This, would add to this paper that can be read and evaluated in clinical practice also by radiologists.

5) FOLLOW-UP: Contrast-enhanced ultrasound should be mentioned as possibility. Please, briefly discuss its potential.

Round 2

Reviewer 3 Report

Comments and Suggestions for Authors

Thank you for your revisions.

The paper is significantly improved.

Please add Panels (a, and b) in Figure 2, since you mentioned them in the legends.

Author Response

Comment 1: [Thank you for your revisions. The paper is significantly improved. Please add Panels (a, and b) in Figure 2, since you mentioned them in the legends].

Reply: Thank you for the comment and for taking the time to go through our manuscript, figure 2b was uploaded to the panel.